# Extracellular Vesicles in Young Serum Contribute to the Restoration of Age-Related Brain Transcriptomes and Cognition in Old Mice

**DOI:** 10.3390/ijms241612550

**Published:** 2023-08-08

**Authors:** Nicholas F. Fitz, Amrita Sahu, Yi Lu, Fabrisia Ambrosio, Iliya Lefterov, Radosveta Koldamova

**Affiliations:** 1Department of Environmental and Occupational Health, University of Pittsburgh, Pittsburgh, PA 15260, USA; 2Department of Physical Medicine and Rehabilitation, University of Pittsburgh, Pittsburgh, PA 15260, USA; 3Department of Physical Medicine and Rehabilitation, Harvard Medical School, Boston, MA 02115, USA; 4Discovery Center for Musculoskeletal Recovery, Schoen Adams Research Institute at Spaulding, Charlestown, MA 02129, USA

**Keywords:** extracellular vesicles, aging, choroid plexus, hippocampus, memory, klotho

## Abstract

We have previously demonstrated that circulating extracellular vesicles (EVs) are essential to the beneficial effect of young serum on the skeletal muscle regenerative cascade. Here, we show that infusions of young serum significantly improve age-associated memory deficits, and that these effects are abolished after serum depletion of EVs. RNA-seq analysis of the choroid plexus demonstrates EV-mediated effects on genes involved in barrier function and trans-barrier transport. Comparing the differentially expressed genes to recently published chronological aging clock genes reveals a reversal of transcriptomic aging in the choroid plexus. Following young serum treatment, the hippocampal transcriptome demonstrates significant upregulation of the anti-aging gene Klotho, along with an abrogated effect after EV depletion. Transcriptomic profiling of Klotho knockout and heterozygous mice shows the downregulation of genes associated with transport, exocytosis, and lipid transport, while upregulated genes are associated with activated microglia. The results of our study indicate the significance of EVs as vehicles to deliver signals from the periphery to the brain and the importance of Klotho in maintaining brain homeostasis.

## 1. Introduction

The conventional wisdom is that a better understanding of the myriad roles of extracellular vesicles (EVs) in CNS homeostasis is essential for developing novel therapeutics to alleviate and reverse the neurological disturbances of aging [1,2,3]. Thus far, an increasing number of studies have revealed the complexities of EV-mediated cell–cell communications in the brain, predominantly emphasizing the role of EV released by brain cells under physiological conditions. It has been well established that EVs can cross brain barriers such as the blood–brain barrier (BBB) and brain–CSF barrier (BCsfB). In fact, the ability of EVs to elegantly pass the BBB and BCsfB has generated interest in exploring EVs and their associated cargo as biomarkers or vehicles with which to deliver drugs and proteins to the CNS. If isolated from CSF and plasma, brain EVs provide an array of options with which to learn about normal brain functions and monitor changes in the brain associated with aging or neurodegenerative pathology [3].

Since humoral factors can enter the brain at the BBB and BCsfB through receptor-mediated transcytosis and micropinocytosis, in recent years, in vivo models, for example, heterochronic parabiosis (HB), heterochronic blood exchange (HBE) [4,5,6,7,8], and infusions of small volumes of plasma or serum, have been used to uncover and better understand the role of those factors in aging and age-related diseases [9,10,11,12]. Recently, using infusions of small volumes of serum, we evaluated the contribution of circulating EVs to the beneficial effect of young serum on aged muscle stem cell (MuSC) function and the skeletal muscle regenerative cascade [6]. We demonstrated that young serum restored a youthful bioenergetic and myogenic profile to old muscle cell progeny and that this effect was dependent on circulating EVs. Yet, the full spectrum of effects of circulating young EVs on aged organs and tissues, including the brain, remains poorly understood. Part of the problem is that brain cell-type-specific EV cargo signatures have not been determined, even in otherwise useful and well-defined in vitro models. Secondarily, the fate of internalized EVs remains largely unknown. While it has been demonstrated that naïve macrophage exosomes cross the BBB and deliver proteins [13], we have an incomplete understanding of the regulatory mechanisms that control the entrance of EVs through the barriers, the effects of EVs delivered through the brain barriers, if and to what extent the effects in response to internalized EV can be ascribed specifically to a certain cell type, and how major brain cell types react to and mediate those effects [14]. Since the internalization of membranous structures depends, at least in part, on their lipid composition, age-dependent changes of the EV lipidomic profiles are important characteristics of the EVs used in rejuvenation studies. Surprisingly, there is a limited number of publications addressing age-dependent differences between EVs isolated from young and aged mice [6,15]. Even more significant is the lack of systematic studies addressing the important question of whether and how EVs isolated from young blood differentially affect cell types in CNS of injected aged mice. Therefore, further in vivo studies are needed in order to establish a reliable and more complete understanding of the role of EVs that are generated in peripheral organs and enter the brain. Additionally, there is a need to investigate along the brain’s response to molecular, cellular, and physiological levels.

In this study, we examined the effect of young serum on the cognitive performance of aged mice. We show that repeated infusions with small volumes of young serum significantly improved age-associated memory deficits and this effect was abrogated after the serum was depleted of circulating EVs. RNA-seq analysis of choroid plexus demonstrated effects on genes involved in barrier function and trans-barrier transport. Interestingly, the hippocampal transcriptome demonstrated a significant upregulation of *Klotho* (*Kl*) gene, which codes for the longevity protein Klotho [16], following young serum treatment. Notably this effect was abrogated after serum EV depletion, suggesting that EVs may serve as vehicles for Klotho messages from the periphery to the brain. Given the well-established role of Klotho in cognitive functioning, we performed transcriptional profiling of *Klotho* knockout (*Kl^ko^*) and *Klotho* heterozygous (*Kl^het^*) mice and found an association with downregulated categories such as transport, exocytosis, and lipid transport, while upregulated genes were associated with microglia. To test if changes in transcriptomes represent transcriptomic rejuvenation, we correlated the transcriptomes of the treated mice to the most recently published chronological aging clocks [17], which revealed a reversal of transcriptomic aging in the choroid plexus. The results of our study indicate that further evaluation of EVs as vehicles for delivering signals from the periphery to the brain and coordinating the responses of different brain regions will open new avenues with which to expand the research and to better understand aging and rejuvenation.

## 2. Results

### 2.1. Circulating EVs Contribute to the Beneficial Effect of Young Serum on Cognitive Functions

Increasing evidence from heterochronic parabiosis studies, blood exchange and plasma transfer models demonstrates that circulating factors in young blood can rejuvenate multiple organs in old mice, including the brain [12]. However, the factors related to these rejuvenating effects remain poorly defined. Plasma transfer experiments, in which aged mice were systemically treated with plasma from young mice, demonstrated improvements in the hippocampal-dependent learning and memory of the aged mice [12]. We hypothesized that EVs in young serum carry factors that facilitate a pro-youthful phenotype. To test this hypothesis, we utilized a previously described [6] experimental paradigm where aged mice (22–24 mo) received tail vein injections of young serum (3–4 mo), young serum depleted of EVs, or saline (sham) for three weeks prior to the start of behavioral testing, as shown in Figure 1A and Appendix A. For both behavioral tests, young serum treatment of aged mice significantly improved cognitive performance compared to the correspondingly aged sham controls (Figure 1B,C). In contrast, when the young serum was depleted of EVs, the beneficial effect of young serum on the cognitive performance of aged mice was abrogated, and mice displayed performances comparable to those of sham controls in terms of both novel object recognition (Figure 1B) and fear conditioning (Figure 1C). The young serum improvement on the memory of aged mice was hippocampal-dependent as there was no significant difference during the amygdala-dependent cued phase (Appendix A). Furthermore, there were no changes during the learning phase or novel phase of fear conditioning, nor was there a significant change in locomotor activity between genotypes (Appendix A). We concluded from the behavioral data that EVs circulating in young serum are essential for the rejuvenation of cognitive function in aged mice.

### 2.2. Aging Shifts Phospholipid Profiles of Circulating Serum EVs

Previously, we have shown that serum EVs from young animals restored skeletal muscle bioenergetics and the regeneration potential of aged mice [6]. This was due in part to an age-related shift in serum EV nucleic acid cargo that affected the skeletal muscle transcript levels of the pro-longevity protein, α-Klotho. In the present study, we further characterized age-related changes in serum EVs that could impact their signaling potential and target cell responses. Lipids are essential molecular components of EVs and act as bioactive molecules, carriers of lipid mediators, and impact binding to recipient cells [3,15]. It has already been demonstrated in multiple studies that the lipidomic profiles and lipid composition of EVs substantially differ in comparison to those of the parent cells [15,18]. To assess changes in the lipid content of serum EVs associated with aging, we applied multi-dimensional mass spectrometry shotgun lipidomics (MDMS-SL) to measure nine major lipid classes and their molecular species. We initially isolated and characterized EVs from the serum of young (3–4 mo) and aged mice (22–23 mo). Nanoparticle tracking analysis (NTA) of concentration and particle size, conducted using the NanoSight instrument, revealed that isolated serum EVs had an average size of 120.8 ± 4.2 nm and concentration of 2.1 × 10^10^ ± 0.33 particles/mL and no difference in EV size or concentration (Figure 2A,B). We also observed no significant difference in the total protein content (Figure 2C) of isolated serum EVs from young and aged mice. NanoSight NTA also confirmed that the EV isolation protocol resulted in the depletion of >95% of serum EVs and an enrichment in EV concentration in the isolated fraction (Figure 2D).

We found that seven of nine phospholipid classes (FA, TAG, PC, PE, CAR, SM and CE; for abbreviations see legend of Figure 2 and Appendix A) had statistically higher levels in EVs isolated from young mouse serum compared to those extracted from aged serum (Figure 2E). Overall, except for PE, young serum EVs had a similar direction of fold change compared to aged EVs for each lipid class assessed (Figure 2F). Additionally, specific subspecies of all lipid classes, except LPC, were found in significantly less abundance in aged serum EVs (Appendix A). The fact that CL, PA, PG, PI and PS classes were undetectable indicates that contamination from mitochondria and ER fractions was minimal [15]. However, as we detected a relatively high TAG content, we cannot exclude contamination with serum lipoproteins in these samples. Thus, our conclusion is that young EVs have a higher phospholipid content than EVs isolated from old sera.

### 2.3. Specific Effect of Young Serum on Gene Expression of Aged Choroid Plexus

Our next goal was to determine how the signal the peripheral injection of young sera is transmitted to the brain and assess the role of EVs in mediating these effects. Therefore, we first focused on the response of the choroid plexus transcriptome as it is a unique barrier situated at the interface of the blood and CSF (BCsfB). We performed RNA-seq on a dissected choroid plexus isolated from the same 3 experimental groups according to the treatment outlined. These groups were sham, young serum (YS), and young depleted serum (YDS), as shown in Figure 1A. To determine differentially expressed genes (DEGs) across the groups and to test the hypothesis that EVs circulating in young serum facilitate transcriptional changes in old mice, we used edgeR and processed the bulk RNA-seq data of the three groups simultaneously (Appendix A). In the choroid plexus, comparing the transcriptomes of mice injected with young serum vs. sham (Figure 3A), or young serum depleted of EVs (Figure 3B), we identified a large number of significantly upregulated genes and a lesser number of downregulated genes at a Log2 fold change > 0.5 cutoff. There were significantly more downregulated genes (129) than upregulated ones (8) when comparing mice injected with YDS vs. sham (Figure 3C), suggesting that the remaining material after the removal of EVs from the young serum has more of a suppressive effect on gene expression. The examination of gene ontology (GO) categories of genes downregulated in CP of mice injected with YDS vs. Sham demonstrated that these are cellular processes involved in innate immune response, suggesting that factors present in young serum after the removal of EVs can suppress these biological processes (Appendix A) [19]. Within the first two comparisons—YS vs. sham and YS vs. YDS—we identified 337 genes that were commonly upregulated in association with the YS treatment and classified as dependent on EVs. By chance, this was greater than predicted, as determined by hypergeometric distribution testing (Figure 3D—dark orange on Venn diagram). We did not identify any common overlapping downregulated genes between the same comparisons (Figure 3D). GO analysis performed using these 337 commonly upregulated genes revealed that top biological processes related to cell adhesion, ion and transmembrane transport, nervous system development and cell migration were upregulated in mice treated using young serum (Figure 3E). Within the top 20 genes of the cell adhesion GO term, we observed the increased expression of many genes that are important to maintaining the integrity of the blood–CSF barrier at the choroid plexus and that are associated with YS treatment (Figure 3F). These upregulated genes included, *Pcdh1*, which encodes a member of the cadherin superfamily and is highly expressed in the choroid plexus epithelial cells [20]. The loss of function of *Pcdh1* results in reduced barrier integrity and increased barrier leakage [21]. The cytoplasmic tails of these cadherins form multiprotein cadherin–catenin complexes. These are the major structural units of adherens junctions and are important in the transport of the choroid plexus [22]. In mice treated with young serum, there was also a significant transcriptional upregulation of a member of this catenin family—*Ctnnd2*. In mice injected with young serum, we also found upregulation of the gene *Vtn*, coding for vitronectin. Studies have shown that binding vitronectin to the α5 integrin receptor instructs brain barrier cells to maintain membrane tension and inhibit transendocytosis, thereby reducing barrier permeability [23,24]. Furthermore, following YS treatment, we observed a significant increase in the number of genes (*Ntm*, *Astn1*, *L1cam*, *Nrxn1*) encoding for adhesion molecules with important roles in neuronal migration, axonal growth, and synaptogenesis. When comparing transcriptomes of mice treated with YS or YDS that is enhanced by the presence of EVs, we also determined changes in expression levels of a significant number of cell-type-specific genes that are upregulated in the aged choroid plexus, including oligodendrocytes (Figure 3G), endothelial cells (Figure 3H), pericytes (Figure 3I) and neurons (Figure 3J). To determine the cell specificity, we used published data on single-cell RNA-seq performed on mouse brains [25,26]. Concomitant with our findings about the upregulation of genes associated with cell adhesion via YS treatment, we also observed increased expression of *Apold1*—a protein known to play a role in the regulation of endothelial cell signaling, particularly of tight junctions [27]. In a pericyte-specific transcriptome, YS treatment increased the expression of *Nbl1*, which promotes choroid plexus-mediated neurogenesis [28]; *Gpc3*, which encodes a protein that complexes with UNC5 to facilitate neural migration in cortex [29]; and *Kcne4*, which modulates vascular tone [30] compared to YDS without EVs. In response to YS administration, we also found an increase in the expression of ceruloplasmin (*Cp*), a ferroxidase enzyme present in CSF which has a neuroprotective role during cerebral ischemia [31] and displays oxidative deamidation in the CSF of Alzheimer’s and Parkinson’s disease patients [32]. In this regard, following YS treatment, we observed an increase in oligodendrocyte-specific genes including *Rap1gap*, which is indicated in controlling cell growth and differentiation, *Sdc3* which is involved in organizing actin cytoskeleton affecting cell shape, and *Adgrl3*, which is important in cell adhesion. Recently, it has been shown that myelin-derived antigens can activate the choroid plexus, which helps regulate the recruitment of immunoregulatory cells to the brain and attenuates neurodegeneration [33]. However, these benefits were abrogated in the absence of EVs from young serum. There were no cell-type-specific downregulated genes in the YS group versus sham or YDS.

During the last 10 years or so, age prediction models, frequently called “aging clocks”, have gained significant popularity [17,34]. In general, the goal of those studies is to build models that can serve as integrative aging biomarkers. Thus, similar to the experimental design seen in the most recent report [17], we asked whether chronological aging clocks, defined by and based on single-cell gene expression levels, capture the rejuvenating effect of YS-associated circulating EVs in old mice. To evaluate the rejuvenating effect of YS infusions and determine if there was a reversal of the chronological aging clock in old mice, we used upregulated genes in the choroid plexus of old mice infused with YS (EVs + circulating factors) compared to YDS (circulating factors but no EVs) (total of 1006 upregulated genes; Figure 3B) and compared them against the 594 downregulated clock genes identified by Buckley et al. [17]. The Venn diagrams (Figure 3K,L) show 52 genes that were downregulated according to the chronological aging clock but reversed—upregulated—in old mice following infusion of YS, which by chance is greater than the number predicted. In terms of the cell specificity of these genes, they were microglia, astrocytes, and endothelial and neuronal precursor cells. GO analysis using these 52 genes revealed that the most affected functional categories were immune system process (*Cd14*, *Slc15a3*, *Csf1r*), regulation of angiogenesis (*Meg3*, *Sema6a*) and ion transport (*Kcna1*, *Slc4a4*, *Kcnk1* and *2*). Among microglial specific genes, genes were mainly associated with homeostatic microglia state (*Siglech*, *Cx3cr1*), whose expression was reported to decrease with aging [35].

We conclude the transcriptional profile of the aged choroid plexus is significantly impacted by 25 days of YS treatment. We see an increased expression of a number of genes that can signal a more youthful phenotype and restore many of the functions of choroid plexus, including barrier establishment, the regulation of CSF, and the maintenance of neural homeostasis. These effects are largely dependent on the presence of EVs in the young serum.

### 2.4. Unique Gene Expression Changes in the Aged Hippocampus of Young Serum-Treated Mice

Previous experiments based on heterochronic parabiosis have shown that when aged animals were exposed to young blood, the effects were ascribed to synaptic plasticity-related transcriptional changes in the hippocampus of aged mice [12]. Unlike in the choroid plexus, following YS treatment, we identified fewer differentially expressed genes in the hippocampus ascribed to EVs (Appendix A). We postulate that the peripheral treatment of YS greatly impacts the choroid plexus compared to the hippocampus based on its location as a unique barrier situated at the blood–cerebrospinal fluid (CSF) interface. Comparing YS vs. sham (Figure 4A), we identified 58 upregulated genes and comparing YS vs. YDS, we identified 75 genes (Figure 4B). We also observed a smaller number of DEGs when comparing the transcriptomes of mice treated with YDS vs. sham, especially in upregulated genes (Figure 4C). As a result of finding relatively few DEGs in both comparisons, there were only 17 (upregulated) and 13 (downregulated) overlapping genes associated with YS treatment when compared to sham or YDS treatment. By chance, this is greater than predicted and we attribute this phenomenon to the presence of circulating EVs (Figure 4D). Of particular interest was the significant upregulation of *Klotho* (*Kl*) following YS treatment and the abrogation of this effect after injection with YDS (Figure 4E). Expression of *Kl* mRNA and protein has been shown to be significantly reduced in the brain with age [36] and is associated with increased microglial activation and inflammasome activation [37]. Aging is also associated with a loss of *Kl* mRNA cargoes in EVs [6]. We also observed increased expression of membrane transporters including: *Abca4*, which is a member of the ATP-binding cassette transporter family and thought to be involved in lipid homeostasis through the transport of phosphatidylethanolamine lipid species; and *Slc39a4*, which encodes a zinc/iron-regulating transporter protein [38] (Figure 4E). Downregulated DEGs in the hippocampus, associated with YS treatment, included genes encoding for neuropeptide vasopressin (*Avp*), tachykinin 2 (*Tac2*), and gamma-synucleins (*Sncg*) (Figure 4F). The Venn diagram (Figure 4G) shows significantly upregulated genes associated with YS treatment (75 genes in total when compared to YDS from Figure 4B) in the hippocampus compared with the chronological clock genes (594 genes in total) downregulated with aging. As visible, there were only 3 overlapping genes, suggesting that the effect of EVs in YS on reversing “aging clock” in hippocampus, at least in this treatment protocol, was minimal.

While we observed fewer transcriptional changes in hippocampus associated with YS treatment, we did see a significant increase in *Kl*, *Abca4*, and *Slc39a4* expression—genes coding for the essential proteins involved in transmembrane transport, again highlighting the neuroprotective potential of EVs and their cargo found in YS.

### 2.5. Klotho Deficiency Significantly Affects Hippocampal Gene Expression

To investigate and further reveal the effect of α-Klotho on the hippocampal transcriptome, we compared hippocampi from 9-month-old Klotho heterozygous (Kl^het^) (Figure 5A and Appendix A, 637 genes downregulated and 905 upregulated in Kl^het^) and 2-month-old α-Klotho knockout (Kl^ko^) mice (Figure 5B, 2264 genes downregulated and 2052 upregulated in Kl^ko^) to their respective age-matched wildtype (WT) controls. Transcriptome analysis was performed via bulk RNA-seq using RNA isolated from dissected hippocampi. Transcriptomic profiling, comparing Klotho-deficient mice to their WT controls, showed a significant number of DEGs at a cutoff of *p* < 0.01 (Figure 5A,B). Gene ontology analysis revealed that terms associated with translation, mitochondrial electron transport, and responses to oxidative stress were upregulated in both comparisons (Figure 5C). The most significant downregulated categories that were common for Kl^ko^ and Kl^het^ were related to transport, exocytosis, lipid transport, and cilium morphogenesis (Figure 5D). Of note, many of the differentially regulated genes in the aforementioned categories code for proteins that are critical for the generation of ectosomes and exosomes, and, therefore, may have a significant impact on the generation and transfer of peripheral signals to the CNS.

Recent studies have identified a subset of microglia named “disease-associated microglia” (DAM) as accumulating in AD patients and other neurodegenerative diseases at the sites of damage [39]. DAM genes are characterized by an upregulation of genes involved in lysosomal, phagocytic, and lipid metabolism pathways, including genes known as AD risk factors, such as *APOE*, *TREM2* and *TYROBP* [40]. It has been observed that, with the progression of neurodegeneration, DAM genes are upregulated, which coincides with a significant downregulation of the so-called “homeostatic microglial” (referred to as “homeo”) genes, such as *P2ry12*, *P2ry13*, *Cx3cr1*, *CD33*, and *Tmem119* [39]. We searched our data sets for up- and downregulated DAM and homeo genes, with the rationale that *Kl* deficiency might participate in encouraging neurodegeneration as well as the upregulation of DAM genes and a decreased expression of homeo genes. Surprisingly, we found that approximately 30% of DAM genes were significantly upregulated in *Kl*-deficient mice (210 out of a total of 650 DAM genes identified in our data set, Figure 5E). Among those were *Apoe*, *Trem2*, *Tyrobp*, *Axl*, *Cd68*, and *Cd74*, as well as many of MHC class II genes (Figure 5E, also shown on volcano plots on Figure 5A,B). Notably, variants of these genes (*APOE* and *TREM2*, *PICALM*, *ADAM10*) have been identified by GWAS studies as associated with risk for late-onset AD [40]. We also found a complementary decrease in 146 homeo genes (out of total 760 identified), such as *Picalm*, *Adam10*, and *Sort1* (Figure 5F). We underscore that the level of upregulation of DAM genes and downregulation of homeo genes differs between those we regularly identify in APP transgenic mice and which are associated with amyloid deposition, suggesting that the neurodegenerative process in *Kl*-deficient mice is different when compared to AD mouse models.

## 3. Discussion

We demonstrate that infusions of YS in aged mice improved their cognitive performance and that this effect was largely dependent on serum EVs. We chose to deplete EVs with the size range of 30–150 nm that are formed via the exocytosis of multivesicular bodies (MVBs), liberating intraluminal vesicles upon fusion with the plasma membrane. This population of EVs provides a snapshot of the parent cells that release them, and upon uptake by the recipient cell, they can modify cellular functions based on EV—associated cargo. These features highlight their importance in cell-to-cell communication. While the methodology of the study does not allow for vesicular tracking, internalization, or the evaluation of responses given by specific cell types, the recorded changes in behavioral phenotypes support the assumption that the underlining reasons for improved cognitive performance were changes in gene expression in neuronal and other cell populations in the hippocampus. The discovery of and research into plasma-soluble factors affecting cognitive performance began more than 40 years ago, when β-endorphin was shown to increase in the circulation with exercise of sufficient intensity and duration [41]. During the last 20 years, multiple studies have demonstrated that the infusion of β-endorphin, other soluble “exercise factors”, as well as small volumes of plasma or serum, sends signals to the brain, increases hippocampal neurogenesis in young mice, rescues age-related neurogenesis and improves cognition in old mice (reviewed in De Miguel et al. [10]). Here, we report cognitive improvement in old mice following 25 days’ treatment using intravenous injections of serum extracted from young animals. Importantly, the improvement in cognitive performance evaluated by two behavioral tests was abolished if the young serum was depleted of EVs. We used RNA-seq and provided lists of genes differentially affected in the hippocampus and choroid plexus of injected old animals in order to test the hypothesis that the presence of EVs within young serum facilitates transcriptional changes, resulting in a rejuvenative aging effect. Prior studies have shown that heterochronic infusion of young CSF into the lateral ventricles of aged mice results in the enhanced proliferation and differentiation of precursor cells in the hippocampus, as well as cognitive improvement [42]. That the choroid plexus plays a key role in transmitting extrinsic signals to brain areas responsible for changes in cognitive performance is one important interpretation of those results. While our study was not designed to reveal the role of specific genes in the choroid plexus or hippocampus on cognitive performance, the results clearly demonstrate that the majority of DEG are in the choroid plexus and that the improvement seen in cognitive performance was significant.

The results of our study demonstrate the differences between the lipidomic profiles of young and aged serum EVs. The differences in the lipid composition of EVs and releasing cells, as well as the enrichment in specific lipid classes and species during the process of EV biogenesis and their release from cells, have been known since EVs were established as biologically and clinically important subcellular structures. Cell-type-specific differences in lipid profiles of EVs also exist [18]. Importantly, EVs isolated in vitro, in vivo or from tissue also differ in their lipid content [15]. While there no common rule or biological process has been identified as responsible for the differences, most of the studies show that EVs are enriched in ceramide, which is associated with EV biogenesis and release; sphingomyelin, which is the source of ceramide; phosphatidylserine; and other glycosphingolipids [43,44]. While differences in lipidomic profiles of EVs isolated from serum or plasma, or in response to certain exposure of healthy individuals or patients have been reported, interpretations of the differences are difficult and speculative [45,46,47]. Differences in lipidomic profiles of EVs isolated from young and aged mice have not been reported so far in the context of a rejuvenating study. In our previous study, we used Raman spectroscopy analysis [6] to detect differences in the lipid content of EVs isolated from young and aged serum and showed an increased lipid content in aged EVs. While it is intrinsically difficult to compare the level of enrichment of EV lipid species isolated from different biofluids and tissues in different studies and different laboratories [15], here, we report differences calculated using MS spectra obtained from EV lipids isolated and processed via the exact same techniques. Therefore, in the absence of lipids indicating contamination from mitochondria, together with the increased levels of lipid species in 7 out of 9 analyzed lipid classes, including ceramide and SM, this study most probably presents a realistic picture of existing difference. However, as noted in the results, we detected a relatively high TAG content. This suggests some contamination with serum lipoproteins in these samples, which is difficult to avoid considering the current methods for EV isolation, and represent one of the limitations of this study. While the widespread opinion is that the lipid composition of EVs is cell-type-dependent, since we isolated serum EVs, our study does not allow for the association of EVs with a certain cell type. At this time, we also cannot speculate as to whether the presence of lipids in the outer leaflet of EV membranes facilitate their internalization in the choroid plexus and endothelial cells of brain microvasculature and their transcytosis. However, based on this research and our previous study, we hypothesize that the cargo of young EVs delivered through BCsfB and BBB to hippocampus causes changes in the cognitive performance of older mice. Further studies are necessary to fully characterize EVs isolated from young mice and profiling of RNAs, including non-coding RNAs and cargo proteins.

During the last 10 years or so, age prediction models, frequently called “aging clocks”, have gained significant popularity [17,34]. In general, the goal of those studies is to build models that can serve as integrative aging biomarkers. While previous molecular aging clocks have largely relied on datasets built using bulk tissue/RNA input or purified cell populations [48,49,50], the most recent study, performed using single-cell sequencing technologies, provided cell-type-specific transcriptomic aging clocks [17]. These new molecular aging clocks allow for the analysis of interventions that aim to counter aging and age-related diseases. The results of the study conducted in Brunet’s lab [17] are high-resolution aging clocks derived from single-cell transcriptomic data. These provide an opportunity to test for and quantify transcriptomic rejuvenation. Most importantly, the cell-type-specific aging clocks derived from mouse SVZ neurogenic regions used in their study generalized to the same cell types in other regions of the brain, and even to other cell types and tissues. While Buckley et al. [17] tested whether the single-cell aging clocks capture known “rejuvenating” interventions—e.g., parabiosis and exercise—our approach tested whether a different rejuvenating intervention—namely, repeated infusions of small volumes of YS in aged mice—fits the model and allows for the identification of genes that are reversed via intervention. We identified 52 genes in the choroid plexus with reversed expression (upregulated in YS versus YDS) and 3 in the hippocampus, which we attribute to the EVs circulating in YS. The vast majority of those genes are associated with inflammation and anti-inflammatory responses in the brain or in general, as well as membrane permeability, cell adhesion and intercellular communications [51,52,53,54]. A prediction about how a transcriptional rejuvenation, which is revealed after roughly a month of treatment with YS, would change behavioral or other phenotypes would have been highly speculative at this time. However, the research into aging clocks published by Buckley et al. [17] and the results presented here support the idea of designing follow-up studies. These could uncover the long-term effects of rejuvenated transcriptomes in not only old rodents, but also in rodent models that correspond to middle-aged individuals in conditions of disease or under extrinsic stressful influences and which require strong anti-inflammatory responses. We believe that an enhanced mechanistic understanding of the potential of circulating EVs to modulate age-related decline in the brain may help to pave the way towards the development of EV-based therapeutic approaches. Understanding the EV-associated cargo that facilitates these beneficial effects will allow for the administration of engineered EVs that are non-immunogenic, biodegradable, biocompatible, and capable of crossing brain barriers. Furthermore, with the development of EV-based therapeutics, the time course of treatment and any associated temporal transcriptional changes will need to be considered.

There are several limitations to our study and conclusions have been made based on behavioral and molecular phenotypes caused by infusions of YS depleted of EVs. Since EVs are transport vehicles loaded with proteins, as well as different types and classes of RNAs and lipids, the question of what exactly we did/did not e deliver to the bloodstream and then the brain of old mice by injecting YS or YDS—young serum depleted of EVs—remain largely unanswered. While the advances in EVs research provide methodological and technical advice on how to characterize serum/plasma EVs, the same unchanged and active serum humoral factors in YS and YDS obviously exist, and their identification requires the use of different approaches. We cannot also exclude the small possibility that the Exoquick depletion protocol is reducing the presence of plasma protein and HDL in the YDS treatment. However, this protocol was chosen due to previous studies conducted with this method showing the recovery of the highest number of particles, with the most consistent distribution of particle size [55]. Lastly, only male mice were examined in the current study. Thus far, few studies have explored sex differences in EVs and their associated cargo. In a cohort of healthy individuals, no differences were observed in the plasma EV protein profiles [56] or total amounts of small RNA and miRNAs [57] associated with sex. So, more research is needed to determine whether there are differences in EVs and their associated cargo in healthy individuals associated with sex, and particularly with healthy aging. The answers to those questions will have fundamental biological implications and, hopefully, therapeutic ones in aging and neurodegeneration research.

## 4. Materials and Methods

### 4.1. Animals

All animal studies were approved by the University of Pittsburgh Institutional Animal Care and Use Committee, which adheres to the guidelines outlined in the Guide for the Care and Use of Laboratory Animals from the United States Department of Health and Human Services. For all experiments, animals were ear-tagged, samples were number-coded, and investigators performing endpoint analyses were blinded to treatment groups. Aged male wildtype (WT, C57BL/6) mice were obtained from the NIA (21–24 months of age) and used for serum treatment studies and the isolation of EVs. Young male WT (C57/BL6) mice used for serum treatment experiments and EV isolation were obtained from Jackson Laboratories. The original Klotho breeders were obtained from MMRCC and UC Davis. The mouse colony was maintained in-house. Klotho knockout (Kl^ko^, 2 months of age), Klotho heterozygous (Kl^het^, 9 months of age) and age-matched WT controls were used for bulk hippocampal RNA-seq. All experimental mice were kept on a 12 h light–dark cycle with ad libitum access to food and water.

### 4.2. Serum Treatments

Mice were anesthetized using isoflurane and blood was collected via cardiac puncture of the right ventricle with 25-gauge 5/8-inch 1 mL needle. Blood was incubated at room temperature for 60 min and centrifuged for 20 min at 16,000× *g*. The serum was aliquoted into 1.5 mL centrifuge tubes and stored at −20 °C. Any sample displaying hemolysis (as evidenced by significant pink/red coloration) was excluded from the experiments. For the young depleted serum treatment, bulk EVs were removed from young serum using ExoQuick (EXOQ5A). Briefly, 63 µL of Exoquick solution was added to 250 µL serum and incubated on ice for 30 min. Samples were centrifuged at 1500× *g* for 30 min at 4 °C, with the EVs pelleted at the bottom of the tube and the supernatant containing the EV-depleted serum used for treatment. The EV pellet was further analyzed using nanoparticle tracking, bulk phospholipid and protein content and MDMS-SL for the purpose of lipid profiling.

Aged WT male mice were randomly assigned to one of the three treatment groups that received tail-vein injections of 100 µL of young serum, 100 µL of young serum depleted of EVs, or sham injections. Animals were injected every three days over a 25-day period during the evening, prior to the start of behavioral testing. Animals received a total of 10 injections, with the 9th and 10th injections administered during behavioral testing. This treatment regimen was chosen based on previous publications showing the ability of young serum and serum from exercised mice to beneficially impact cognitive functions in aged animals [10,11,12]. We tested four serum samples for platelet contamination using a hemoanalyzer 264 (Hemvet 950FS) and observed minimal to no presence of platelets in the samples (average of 5 K/µL serum).

### 4.3. Extracellular Vesicle Characterization

For Western blotting, equal amounts of whole young serum, young serum depleted of EVs and isolated EVs using the Exoquick as described above were resolved on 4–12% Bis-Tris gels (Invitrogen, Carlsbad, CA, USA) and transferred onto nitrocellulose membranes (ThermoFisher, Waltham, MA, USA, iBlot2 Gel system). These membranes were probed with anti-CD63 and anti-CD81 (System Biosciences, Palo Alto, CA, USA, 1:500 dilution, secondary antibody—goat anti-rabbit HRP, 1:10,000). Immunoreactive signals were visualized using enhanced chemiluminescence with the Amersham Imager 600 (GE Lifescience, Chicago, IL, USA).

Nanoparticle tracking analysis (NTA) was performed using an NS300 NanoSight device 287 (Malvern Panalytical, Malvern, UK). Ten microliters from each EV sample were diluted 1:100 in particle-free water and run through the flow-cell. Three 60 s videos were recorded for each sample, with the camera level set to 9 and manual advance notification. These videos were batch-analyzed with the software (NTA 3.3) with the detection threshold set to 7 for analysis of EV size and particle concentration. The flow-cell was washed with 10 mL of particle-free water between each sample.

### 4.4. Multi-Dimensional Mass Spectrometry Shotgun Lipidomics (MDMS-SL)

The MDMS-SL assay [26,58,59] was performed to determine differences in lipid composition of EVs isolated from young and old WT mice using the ExoQuick protocol described above. Quantitative analysis was performed on a triple-quadruple mass spectrometer (Thermo Fisher Scientific, Waltham, MA, USA) equipped with an automated nanospray apparatus NanoMate and Xcalibur system. Internal standards for quantification of individual molecular species of the major lipid classes were added to each sample prior to extraction. Lipid extraction was performed via the utilization of the methyl-tert-butyl ether (MTBE) method with resuspension in a chloroform/methanol (1:1 *v*/*v*) solution with nitrogen flush. The identification and quantification of all reported lipid molecular species scans from the mass spectrometer was automatically performed with a customized sequence subroutine operated under Xcalibur software (v4.3, ThermoFisher) [58,59]. The resulting MDMS-SL data were normalized to total protein content, which was assessed according to the Pierce BCA Protein Assay Kit (ThermoFisher). Abbreviations used for the classes assessed include: FA: fatty acyl chains in TAG, TAG: triacylglycerol; PC: phosphatidylcholine; PE: phosphatidylethanolamine; CAR: carnitine and acetyl carnitine; SM: sphingomyelin; CE: ceramide.

### 4.5. Behavioral Testing

For three successive days prior to behavioral testing, mice were placed in individual containers and handled for 3 min in the behavioral testing room by the same examiner before being returned to their housing cages. Novel object recognition (NOR) was performed as previously described [26,60]. On Day 1, the habituation phase, each animal was allowed to freely explore an open arena (40 cm × 40 cm × 30 cm tall white plastic box) for two 5 min trials with a 5 min inter-trial interval. On Day 2, the familiarization phase, each animal was returned to the same open arena with two identical objects (multicolored tower of LEGO^®^ bricks 8 cm × 3.2 cm) located in opposite diagonal corners of the arena for the same two trials. On Day 3, following a 24 h retention period, the animal was returned to the open arena and allowed to explore for one 10 min interval with two objects in the same positions as the previous day, but with one object replaced with a novel object (metal bolt and nut of similar size). Mice were placed into the middle of the arena facing the posterior wall to prevent any object preference. The arena and objects were cleaned with 70% ethanol and distilled water between different animals. The activity was recorded and scored with ANY-maze software version 6.3 (Stoelting Co., Wood Dale, IL, USA). The total distance traveled by each mouse was recorded during the habituation phase to assess locomotor activity. Exploration was defined by the software as the mouse sniffing, climbing on, or interacting with an object while facing it within 3 cm. The percentage exploration was determined by dividing the time exploring the novel object by the total time exploring both objects. This calculated value provides an indicator of recognition memory, with less time spent exploring the novel object signifying memory deficits.

Contextual and Cued Fear Conditioning (CCFC) was performed as previously described [26,60]. On Day 1, training phase, mice were placed in a conditioning chamber (Stoelting Co.) for 5.5 min. The first 2 min were silent, allowing for acclimation to the chamber; they were followed by a 30 s tone (2800 Hz; Intensity 85 dB, conditioned stimulus (CS)) that ended in a 2 s foot shock (0.7 mA, unconditioned stimulus (US)) through the bars on the floor. This testing was repeated and ended with 30 s of re-acclimation. On Day 2, the contextual phase, mice were placed in the same conditioning chamber for 5 min with no tone or shock administered in order to measure contextual fear conditioning. On Day 3, the gray walls of the chamber were replaced with black and white striped walls to introduce a novel environment for assessing cued fear conditioning. After the first 2 min of silence, the tone was administered for 3 min, and freezing was determined to assess cued fear conditioning. Freezing time was defined as the absence of movement, except for respiration, and was calculated as percentage freezing of the total time in the chamber during each phase of testing with ANY-maze software (Stoelting Co.).

### 4.6. Animal Tissue Processing

Mice were anesthetized via intraperitoneal injection using Avertin (250 mg/kg of body weight). Blood was collected from the right ventricle, followed by trans-cardiac perfusion through the left ventricle with 20 mL of cold 0.1 M phosphate-buffered saline (PBS), pH 7.4. The brain was removed and divided into hemispheres. One hemisphere was dissected into the hippocampus and choroid plexus and flash-frozen on dry ice for bulk RNA-seq.

### 4.7. Bulk mRNA-seq Data

RNA was isolated from the hippocampus and choroid plexus using the RNeasy mini kit (Qiagen, Germany) and RNA quality was assessed using the 2100 Bioanalyzer (Agilent Technologies, Santa Clara, CA, USA). Samples (RIN > 8) were used for library generation and sequenced on Illumina NovaSeq PE 150 (library generation and sequencing were performed at (Novogene Co. Inc., Singapore). The analysis was performed as in our previous publications [26,60,61,62]. Following initial processing and quality control, sequencing data were aligned to the mouse genome mm10 using Subread (v1.5.3; https://sourceforge.net/projects/subread/files/subread-1.5.3/ (accessed on 25 February 2023)), with an average read depth of 50 million successfully aligned reads. Statistical analysis was carried out using Rsubread (v1.34.2; https://bioconductor.org/packages/release/bioc/html/Rsubread.html (accessed on 25 February 2023)), DEseq2 (1.24.0; https://bioconductor.org/packages/release/bioc/html/DESeq2.html (accessed on 25 February 2023)), and edgeR (v3.26.5; https://bioconductor.org/packages/release/bioc/html/edgeR.html (accessed on 25 February 2023)), in R environment (v3.6.0; https://www.r-project.org/ (accessed on 25 February 2023)). Specifically, library sizes were normalized with the “calcNormFactors” function in edgeR, and the differential expression between two groups were performed using the functions “glmQLFit” and “glmQLFTest”. Functional annotation clustering was performed using the Database for Annotation, Visualization and Integrated Discovery (DAVID; https://david.ncifcrf.gov (accessed on 25 February 2023)). Hypergeometric distribution testing for commonly expressed genes between comparisons was performed using phyper in an R environment.

### 4.8. Statistical Analysis

Sample sizes (n) for number of samples per treatment group is indicated in the figure legends. No outliers were removed from the analysis. All results are reported as means ± SEM. Behavioral data were analyzed via one-way ANOVA followed by Tukey’s multiple comparison test. Characterizations of EVs, including the MDMS-SL, were analyzed via two-tailed unpaired *t* test. All statistical analyses were performed in GraphPad Prism (v 9.4.1), or R (v 3.6.0) and significance was determined as *p* < 0.05.

## Figures and Tables

**Figure 1 ijms-24-12550-f001:**
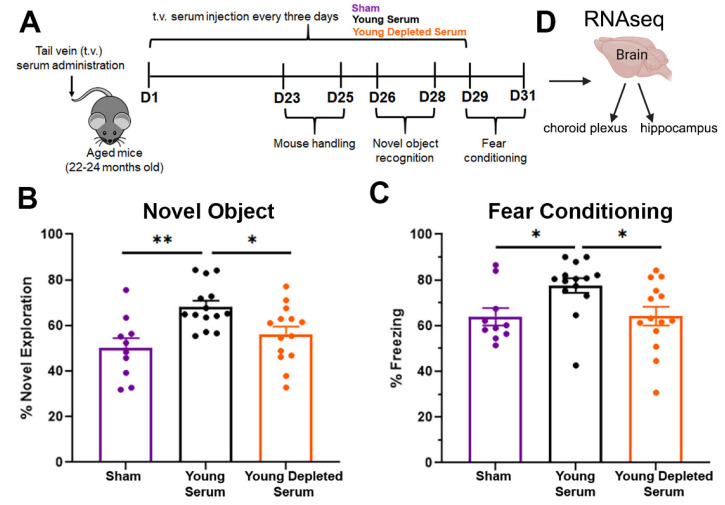
**Beneficial effect of young serum on cognitive performance of aged mice is dependent on circulating extracellular vesicles (EVs).** Experimental paradigm schematic (**A**) for assessing changes in cognitive function of aged mice (22–24-months-old) after treatment with either: sham (saline), young serum or young serum depleted of EVs. Bar plots depicting the performance during novel object recognition (**B**) and contextual cued fear conditioning (**C**) of mice from the three treatment groups. Analysis using one-way ANOVA, followed by Tukey’s multiple comparison test. Bars represent mean ± SEM. n = 10 for sham; n = 14 for Young Serum and Young Depleted Serum with equal sex distribution. * *p* < 0.05, ** *p* < 0.01. (**D**) After behavior the mice were perfused and brain removed. Choroid plexus and hippocampus were dissected and used for RNA-seq.

**Figure 2 ijms-24-12550-f002:**
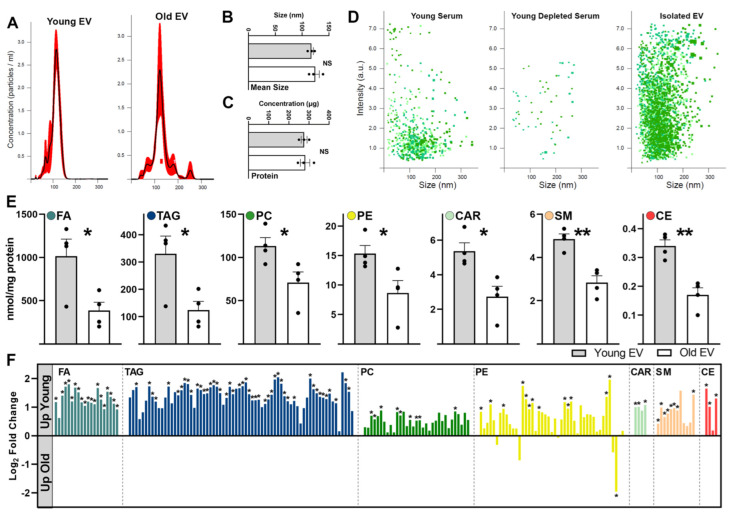
**Comprehensive analysis of mouse serum EVs identifies age—associated changes in lipid profiles.** Representative histograms (**A**) depicting particle size distribution determined by NanoSight-based nanoparticle tracking analysis of young and aged mouse serum EVs. Black lines represent the mean with red denoting 95% confidence intervals. Bar plots summarizing mean size (**B**) or concentration (**C**) of isolated serum EVs. Scatter plots (**D**) depicting particle size distribution determined by NanoSight of young serum, young serum depleted of EVs and isolated EVs utilizing ExoQuick protocol. Bar plots (**E**) depicting major lipid class composition of isolated serum EVs from young and old mice, as determined using MDMS-SL. Bar charts (**F**) summarizing fold change between young and old serum EVs for all major lipids species from significantly changed lipid classes. FA: fatty acyl chains in sample; TAG: triacylglycerol; PC: phosphatidylcholine; PE: phosphatidylethanolamine; CAR: carnitine and acetyl carnitine; SM: sphingomyelin; CE: ceramide. NS, not significant. Analysis using two-tailed unpaired *t* test. Bars represent mean ± SEM. n = 4 for both young and old serum EVs with equal sex distribution. * *p* < 0.05, ** *p* < 0.01.

**Figure 3 ijms-24-12550-f003:**
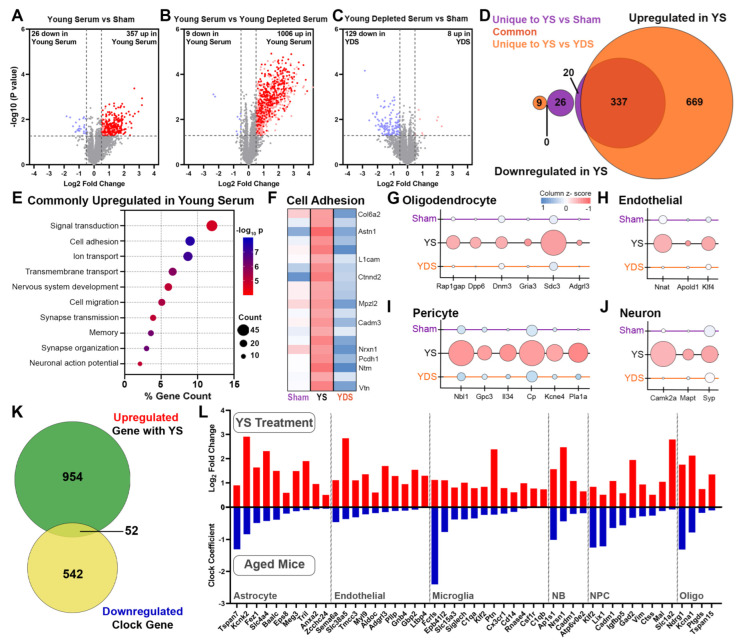
**Specific effect of young serum treatment on gene expression of aged choroid plexus.** Gene expression profiling was performed using RNA-seq on samples of dissected choroid plexus from the same 22–24-month old mice that underwent behavioral testing (Figure 1) and edgeR was utilized to identify differentially expressed genes between treatment groups. Volcano plots represent differentially expressed genes between Young Serum vs. Sham (**A**), Young Serum vs. Young Depleted Serum (**B**), and Young Depleted Serum vs. Sham (**C**). Shown are genes at *p* < 0.05 and Log2 FC > 0.5 cutoffs, with dark red dots denoting commonly upregulated genes in Young Serum when compared to either Sham or Young Depleted Serum. Venn diagram (**D**) represents both significantly up- and downregulated genes in Young Serum vs. Sham and Young Serum (YS) vs. Young Depleted Serum (YDS) comparisons with 337 genes that were commonly upregulated in association with YS, which is greater than predicted by chance as determined using hypergeometric distribution testing (*p* < 0.05). Bubble plots (**E**) depicting GO terms associated with the 338 commonly upregulated genes in YS. *X* axis indicates % gene count, circle size is positively correlated with gene number, and circle color denotes statistical significance. Heat map (**F**) showing the expression level of the top 20 genes that compose the Cell Adhesion GO term. Bubble plots of genes commonly upregulated in YS group specific to oligodendrocytes (**G**), barrier-associated cells (**H**), pericytes (**I**) and neurons (**J**). Size of the circle is positively associated with gene expression and color indicates Z-score. Venn diagram (**K**) showing comparison of significantly upregulated genes associated with YS in the choroid plexus (1006 genes, Figure 2B) compared to published chronological clock genes (594 genes) downregulated with aging, with 52 common genes (hypergeometric distribution testing, *p* < 0.05). Bar plots (**L**) depicting the 52 genes that showed upregulation following young serum treatment and downregulating with aging, organized by cell type (NB: neuroblast; NPC: neural progenitor cells; Oligo: oligodendrocytes). n = 8 for sham; n = 12 for young serum; and n = 14 for young depleted serum with equal sex distribution.

**Figure 4 ijms-24-12550-f004:**
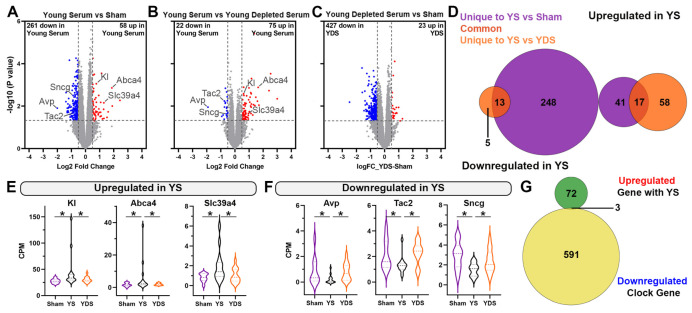
**Unique gene expression changes in the aged hippocampus of young serum—treated mice.** Gene expression profiling was performed using RNA-seq on samples of dissected hippocampus from the same 22–24-month-old mice in Figure 1 and Figure 3 and edgeR utilized to identify differentially expressed genes between groups. Volcano plots represent differentially expressed hippocampal genes between Young Serum vs. Sham (**A**), Young Serum vs. Young Depleted Serum (**B**), and Young Depleted Serum vs. Sham (**C**). Shown are genes at *p* < 0.05 and Log2 FC > 0.5 cutoff. Venn diagram (**D**) was generated from both significantly up- and downregulated genes in the Young Serum vs. Sham and Young Serum (YS) vs. Young Depleted Serum (YDS) comparisons. Violin plots showing the expression level of significantly upregulated (**E**) and downregulated (**F**) DEGs in the hippocampus of Young Serum-treated mice. Venn diagram (**G**) showing the comparison of significantly upregulated genes associated with young serum treatment in the hippocampus compared to published chronological clock genes (594 genes in total) downregulated with aging. n = 8 for sham; n = 12 for Young Serum and n = 14 for Young Depleted Serum with equal sex distribution. * *p* < 0.05.

**Figure 5 ijms-24-12550-f005:**
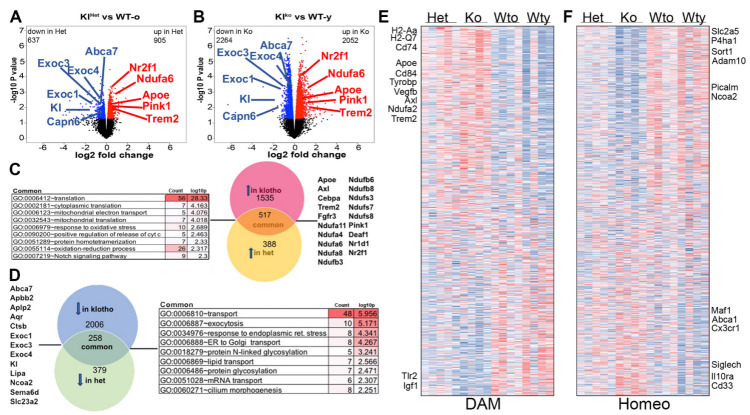
**Effect of Klotho deficiency on gene expression in the hippocampus.** RNA–seq was performed on samples of dissected hippocampus from 2-month-old Kl^ko^, 9-month-old Kl^het^ and age-matched wild-type controls with both young (WT-y) or old (WT-o) serum extracts, and the results were analyzed using edgeR. Volcano plots represent the differentially expressed genes, which are colored in blue (downregulated) and red (upregulated) at *p* < 0.05 for comparisons between Kl^het^ vs. WT-o (**A**); 637 genes downregulated and 905 upregulated and Kl^ko^ vs. WT-y (**B**); 2264 genes downregulated and 2052 upregulated. Venn diagram with expanded GO terms generated from the common genes that significantly upregulated (**C**) or downregulated (**D**) in both comparisons. Heatmap of differentially expressed DAM, with 30% significantly upregulated in Kl-deficient mice (210/650 DAM genes) (**E**). Heatmap of differentially expressed homeostatic microglia genes with 146 downregulated (760 total) in Kl-deficient mice from the hippocampus of the 4 experimental groups (**F**). n = 4–5/group.

## Data Availability

The data that support the findings of this study are openly available in NCBI Gene Expression Omnibus (GEO) database at https://www.ncbi.nlm.nih.gov/geo/query/acc.cgi?acc=GSE234667 (accessed on 12 June 2023), reference number GSE234667.

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
