# Peer review of "Extracellular Vesicles in Young Serum Contribute to the Restoration of Age-Related Brain Transcriptomes and Cognition in Old Mice"

_ijms, 2023, doi:10.3390/ijms241612550_

Round 1
Reviewer 1 Report
The work on 'Extracellular vesicles in young serum contribute to restoration of age-related brain trasncriptomes and cognition in old mice' is novel and interesting but I have concerns about the techniques used for EV isolation and depletion.
1. EVs isolated with Polymer based reagents is known to be associated with low sample purity and potentially affects EV functionality. The contaminants co-isolated with EVs can interfere with Omics analysis. What is the reason behind choosing this method for EV isolation?. Please add this as one of the limitations.
2. There is no proven method that can completely deplete EVs in serum/plasma. Did authors try ultracentrifugation to deplete EVs?
3. Only males were used in the study is there a rationale behind it?
Minor proofreading required
Reviewer 2 Report
The manuscript by Fitz and colleagues explores the effects of extracellular vesicles derived from plasma from young mice on the cognition of old mice, and gene modulation in the hippocampus and choroid plexus. This paper is clear and well-written.
1- What are the consequences of injecting plasma from older mice? The researchers showed results of saline, serum from young mice, or serum from young mice without extracellular vesicles injections. But with the results shown it is not possible to conclude that young serum is essential for the rejuvenation of cognitive function if they don't evaluate the effects of serum from aged mice too.
2 - Please discuss the fact that ExoQuick isolates exosomes (40-100 nm) and not all sizes of extracellular vesicles (30 nm-10 um).
3 - How the other durations of treatment would affect the outcomes that you show? For instance, would you see less gene modulation after 2 weeks of serum treatment, and more genes after 2 months of treatment?
4 - What are the possible reasons for the difference in gene modulation between the choroid plexus and hippocampus? Is there any justification based on location and function? And if most genes are associated with inflammation and permeability, cell adhesion and intercellular communication, is there another way to stimulate rejuvenation using fabricated membranes?
